



# The World Atlas of Last Interglacial Shorelines (Version 1.0)

Alessio Rovere[1,2], Deirdre D. Ryan[3], Matteo Vacchi[3], Andrea Dutton[4], Alexander R. Simms[5], Colin V. Murray-Wallace[6]

[1] DAIS, Ca' Foscari University of Venice (IT)
[2] MARUM, Center for Marine Environmental Sciences, University of Bremen (DE)
    [3] Dipartimento di Scienze della Terra, University of Pisa (IT)
    [4] Department of geoscience, University of Wisconsin-Madison (US)
    [5] Department of Earth Science, University of California, Santa Barbara (US)
    [6] School of Earth, Atmospheric and Life Sciences, University of Wollongong, NSW, 2522 (AU)

*Correspondence to*: Alessio Rovere (alessio.rovere@unive.it)

**Abstract.** In this manuscript, we present Version 1.0 of the World Atlas of Last Interglacial Shorelines (WALIS), a global database of sea-level proxies and samples dated to Marine Isotope Stage 5 (∼80 to 130 ka). The database includes a series of datasets compiled in the framework of a Special Issue published in this journal

(https://essd.copernicus.org/articles/special_issue1055.html). This manuscript collates the individual contributions (archived in Zenodo, https://zenodo.org/communities/walis_database/) into an open-access, standalone database (Rovere et al., 2022, https://doi.org/10.5281/zenodo.6623428). The release of WALIS 1.0 includes complete documentation and scripts to download, analyze, and visualize the data (https://alerovere.github.io/WALIS/). The database contains 4545 sea-level proxies, 4110 dated samples and 280 other time constraints, interconnected with several tables containing accessory data and

metadata. By creating a centralized database of sea level proxy data for the Last Interglacial, the WALIS database will be a valuable resource to the wider paleoclimate community to facilitate data-model integration and intercomparisons, assessments of sea level reconstructions between different studies and different regions, as well as comparisons between past sea level history and other paleoclimate proxy data.

## 1 Introduction

The survey, interpretation, and dating of sea-level index points is an essential tool to assess long-term changes in relative sea level (RSL), which is mediated by the interplay of land motion and global mean sea-level changes. Since the end of the 19th century, scientists have systematically observed and measured archaeological and geological evidence of RSL changes (Celsius, 1743; Issel, 1883; Forbes, 1829). In the 20th century, the observation of past sea-level changes evolved into a standalone discipline, at the crossroads between geomorphology (measurement and interpretation of sea-level proxies, e.g.,

Shennan, 1982), geochemistry (dating of sea-level proxies, e.g., De Vries and Barendsen, 1954), geophysics (modeling





vertical land motions following ice melting, e.g., Peltier, 1974; Farrell and Clark, 1976) and structural geology (understanding the vertical displacement of shoreline features through time, e.g., Chappell et al., 1996).

The growing knowledge on geological sea-level proxies in different regions and the emergence of new and more refined

scientific questions on past RSL changes (see overviews by Murray-Wallace and Woodroffe, 2014, and Gehrels and Shennan, 2015) presents a need for standardization of geological sea-level information. Answers to such need were first presented in a seminal paper by Shennan (1982) and, only a few years later, were explicit in the volume "*Sea-level research: A manual for the collection and evaluation of data*" (Van de Plassche, 1986). After these efforts, approaches to standardize Holocene sea-level proxies were widely accepted by the community working on paleo sea-level changes and have been

recently summarized in an update of the 1986 manual (Shennan et al., 2015).

The community working on past RSL changes has often organized itself around projects supported by the IGCP (International Geoscience Programme, formerly International Geological Correlation Programme) or focus groups supported by the International Union for Quaternary Research (INQUA). One of such groups, the "*PALeo constraints on SEA level

rise*" (PALSEA), founded and currently co-funded by PAGES (Past Global Changes), has advanced  the state of the art on paleo ice sheets and sea levels (Rovere and Dutton, 2021) in the last decade. Results from PALSEA have laid the foundations for improvements in the standardization of paleo sea-level data: creating global sea-level databases, compiled by several experts following a unique database template (Düsterhus et al., 2016). The first multi-proxy, global sea-level database stemming from these efforts compiled published sea-level index points formed after the Last Glacial Maximum (0-

20 ka) and presented as part of a Special Issue in the journal "*Quaternary Science Reviews*" (Khan et al., 2019). The structure used by this database was built upon the concepts already explored within a database template that is widely employed in Holocene sea-level studies (e.g., Engelhart and Horton, 2012). These efforts led to compile a global standardized compilation of Holocene sea-level proxies, including data reviewed by 30 studies (see Table 1 of Khan et al., 2019) and containing a total of 5290 sea-level indicators scattered across the globe.


While studies of RSL changes for periods older than the Holocene have a long tradition (De Lamothe, 1911; Issel, 1914; Blanc, 1936), the standardization of sea-level proxies dating beyond 20 ka into databases has lagged. One significant epoch concerning former sea-level changes is the Last Interglacial (MIS 5, c.130-80 ka). The warmest peak of this interglacial, MIS 5e (c.128-116 ka), is commonly considered a useful (however imperfect) analogue for a future warmer climate, and hundreds

of studies since the early 1900s have described the stratigraphy, elevation, and age of Last Interglacial RSL proxies at thousands of locations.



The present undertaking within this special issue presents a global database reporting geological sea-level proxies formed during MIS 5. We named the database "*World Atlas of Last Interglacial Shorelines*" (WALIS). The name chosen is an homage to the "*World Atlas of Holocene sea-level changes*" by Pirazzoli and Pluet (1991), which, albeit not strictly standardized, was the first attempt at making a global compilation of paleo sea-level data. The database was built collating the individual contributions in the WALIS Special Issue (published by this journal) into a unique database (WALIS 1.0, i.e., Version 1.0). Here, we describe the database structure and the tools associated with it (e.g., interface for inserting data, data analysis tools, and visualization). We also summarize the contents of WALIS 1.0 and the main limitations of the database in its present form.

## 2 Relationship with previous compilations

WALIS is not the first attempt to collect MIS 5 sea-level index points into a coherent database. Databases compiling single studies of MIS 5 sea-level proxies exist for several regions and were built to collate single studies in regional databases (e.g., Ferranti et al., 2006; Ota and Omura, 1991; Muhs et al., 2003, 2002). These summarize the works done over several decades in one region, nation, or wide geographic area, often reporting the results of earlier works that are hard to retrieve in digital format. Regional compilations are usually done by geoscientists with direct expertise on at least some of the sites reported, which increases the confidence that the data have been screened and standardized with a certain degree of knowledge of local geological contexts. In the WALIS ESSD Special Issue, we tried to maintain this advantage by soliciting regional compilations to authors with direct expertise on each area addressed in the database, or with expertise with a particular type of indicator (e.g., coral sea-level proxies, or speleothems) or dating technique (e.g., U-series).

Within the existing literature, several global databases predate WALIS 1.0. One of the most comprehensive in terms of the sheer number of data points was assembled by Pedoja et al. (2011), later updated by Pedoja et al., 2014. In parallel, the works of Dutton and Lambeck (2012), Medina-Elizalde (2013), and Hibbert et al. (2016) contain global compilations of Quaternary U-series data and associated RSL information. Similarly, scientists working on amino acid racemization dating have been working towards a standardized database of dated samples (Wehmiller and Pellerito, 2015). A global compilation of deposits dated with luminescence exists (Lamothe, 2016), albeit the published version contains relatively little metadata (e.g., region of occurrence, luminescence method, sedimentary facies, age, and reference). These databases were used as a starting point for WALIS 1.0. In particular, all the references cited by Pedoja et al. (2014) were initially added to the "References" table and subsequently updated and implemented with new ones by the data compilers working on regional databases within the Special Issue. Hibbert et al. (2016) was the foundation upon which the U-series coral compilation was done (Chutcharavan and Dutton, 2021a). The global compendium of MIS 5a and MIS 5c data (Thompson and Creveling, 2021b) is based on an expansion of a previous compilation included as a table in Creveling et al. (2017). Data from some of



the regional databases mentioned above (e.g., Ferranti et al., 2006; Ota and Omura, 1991) has been re-evaluated and inserted

into the WALIS structure (Cerrone et al., 2021b; Tam and Yokoyama, 2021).

The key difference between previous compilations and WALIS is that the latter is structured as a relational database, with links between records in different tables. Previously published databases have, in general, a simpler structure than WALIS. For example, the databases of Pedoja et al. (2014) and Ferranti et al. (2006) contain several fields describing sea-level index

points, including fields describing the dating methods used and the associated age(s). However, no detailed information is available on the ages themselves (e.g., the minimum fields considered necessary to describe a U-series age by Dutton et al., 2017). In WALIS, such data is instead present in a dedicated table and linked to the table containing sea-level information. Conversely, in the global compilation of coral sea-level index points by Hibbert et al. (2016), the fields to describe U-series ages are present. Still, those related to sea-level stratigraphic details are limited. Instead, these fields are present in WALIS in

the "RSL Stratigraphy" table. As described below, all tables in WALIS are linked with one-to-many or many-to-many relationships.

### 3 The WALIS data platform

Overall, the WALIS database was built alongside a data platform with the same name. Critical parts of the platform are two MySQL databases (one private and one public), a PHP (Hypertext Preprocessor) interface, and a series of visualization, data

download, and documentation tools (see Figure 1 for a schematic representation and Table 1 for links to resources). WALIS was structured around the four key types of scientists interacting with a database, described by Düsterhus et al. (2016). **Data creators** are scientists who produce new data, such as sea-level proxies or radiometric dates, and publish them in research outputs (e.g., peer-reviewed articles or Ph.D. theses). **Data compilers** review these outputs and include the data in the standardized WALIS format. For WALIS 1.0, data compilers are the authors of review manuscripts included in the ESSD

Special Issue (with few exceptions discussed in section 4 Database contents). In some instances, a data creator may coincide with the data compiler. **End users** query and analyze the database for their research. **Database administrators** ensure that data entered by data compilers meets the minimum standards and manage the platform's administration, including fixing bugs and updates. For WALIS 1.0, the database administrators are the authors of this manuscript.

Data compilers can sign up to the interface, which is publicly available and free of charge (Table 1). The data required to sign up are a user-selected combination of username and password (the latter is encrypted in the private database and deleted upon publication in the public database), a valid email address, name, surname, institution, and country. The data compiler must accept a privacy policy (that was built with the Privacy Policy Generator of the German Association for Data Protection). In general, every table in WALIS contains a "Public" field of type integer. For every new data point inserted in

WALIS, this field is set to "0", which means that the record is not publicly visible and can only be seen, edited, and deleted

by the user who inserted it. Exceptions to this rule are tables containing positioning and elevation measurement methods and bibliographic references (Figure 2). Data inserted in these tables is made immediately public and can be seen by all data compilers (and they can be deleted only by Admin). This is set up to ensure that standard metadata (e.g., survey methods) can be used at large and are not duplicated in the database.


All data inserted by a data compiler is embedded within a private MySQL database. This database can be accessed only by the administrators, who periodically revise the non-public records and contact the data compiler to make them public. Upon agreement, the data compiler is invited to upload their data to the WALIS Community in Zenodo (or any other open-access repository) under a Creative Commons Attribution (CC BY) license. Alternatively, the data compiler is requested to give

consent to publish the data in WALIS under this license. At this stage, "Public" fields are changed to "1", and they can only be edited or deleted by submitting a data modification or deletion request to the database administrators. This can be done via the database interface so that post-publication changes are tracked in future WALIS versions.

All data where the Public field is set to "1" are periodically migrated to the public MySQL database, where a general user is

granted "SELECT" rights. End users can download this database or perform SQL queries directly on the public online version (Table 1). For end-users familiar with Python, a series of Jupyter notebooks can be used to perform simple database queries (Table 1). These scripts are also available in GitHub, where they can be forked (i.e., copied and modified without affecting the original repository) and commented on by anyone with a GitHub account. For end-users who do not wish to use SQL or Python, the database is also available for download as a multi-sheet spreadsheet, comma-separated text files, or as

GeoJson files (Table 1).

A combination of Python and R scripts is then run periodically by the database administrators on the public MySQL database. These scripts summarize the data and compile an R ShinyApp for data visualization and download (Garzón and Rovere, 2021). Database administrators also update the database documentation (Rovere et al., 2020) and a series of video

tutorials to guide data compilation. The documentation files are included in a ReadTheDocs website and are also available via GitHub. Users may either fork the original documentation (written in markup language) or suggest improvements via the GitHub "issues" function.

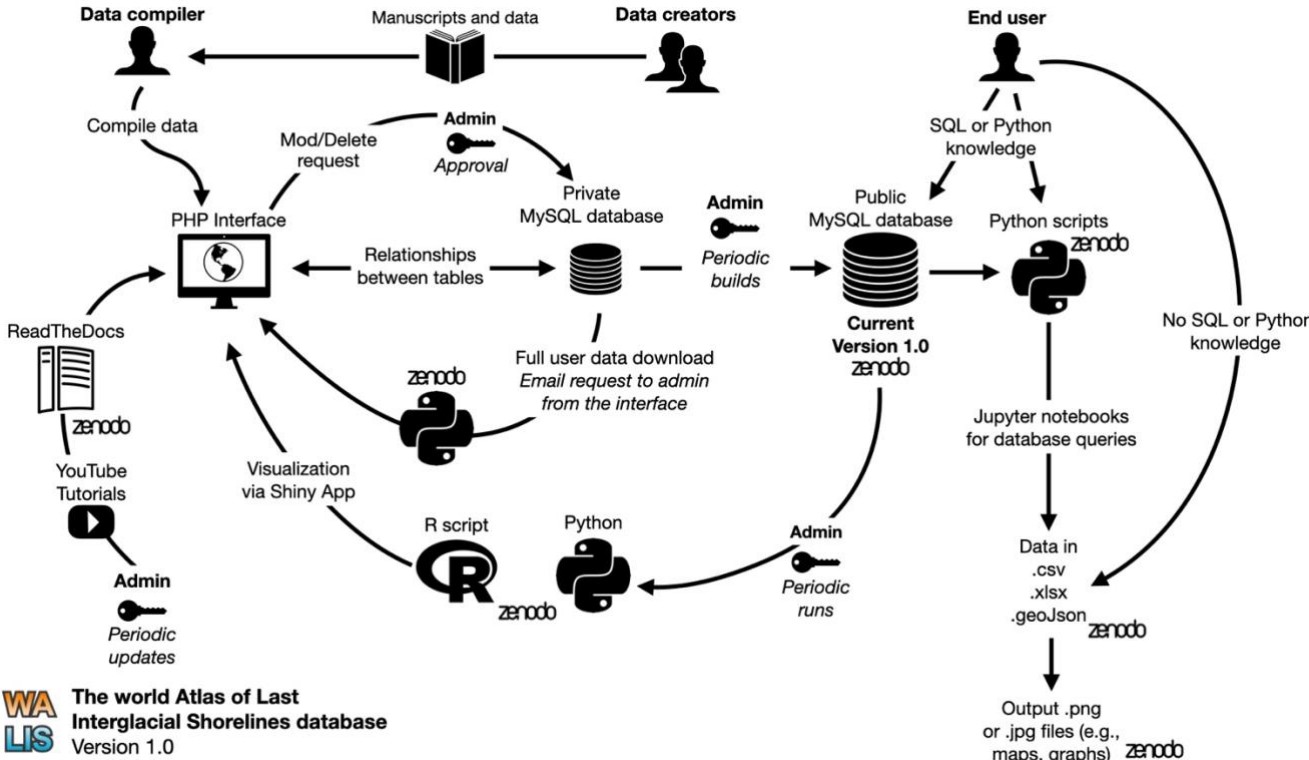

**Figure 1 Overview of the WALIS platform, which includes the private and public instances of the database, the PHP data interface, and scripts in R and Python to query and visualize the data. The "Zenodo" logo indicates that the scripts/data are available in Zenodo (see Table 1 for links).**

### 3.1 The WALIS Database

WALIS is built as a series of tables embedded in a MySQL database. The central concept is that the description of a single sea-level index point must include data and metadata describing its stratigraphy and relationship with former sea level and must be tied to a series of tables reporting on samples and their dating. The techniques used to measure elevation (including the adopted sea-level datum) and geographic positioning must be connected with sea-level index points and dated samples. Also, studies reporting details on each site, sample, or technique used must be included where necessary (Figure 2). The

WALIS database structure (i.e., the mandatory or optional columns that need to be filled when describing a given entity) was created by several scientists (coordinated by the authors of this manuscript) including experts on different subfields (e.g., coastal stratigraphy, dating techniques, earth modeling). A complete list of tables and fields, and associated descriptors, is available via the project documentation (Rovere et al., 2020). An overview of the database in terms of the number of fields and records contained in each table is shown in Figure 2.


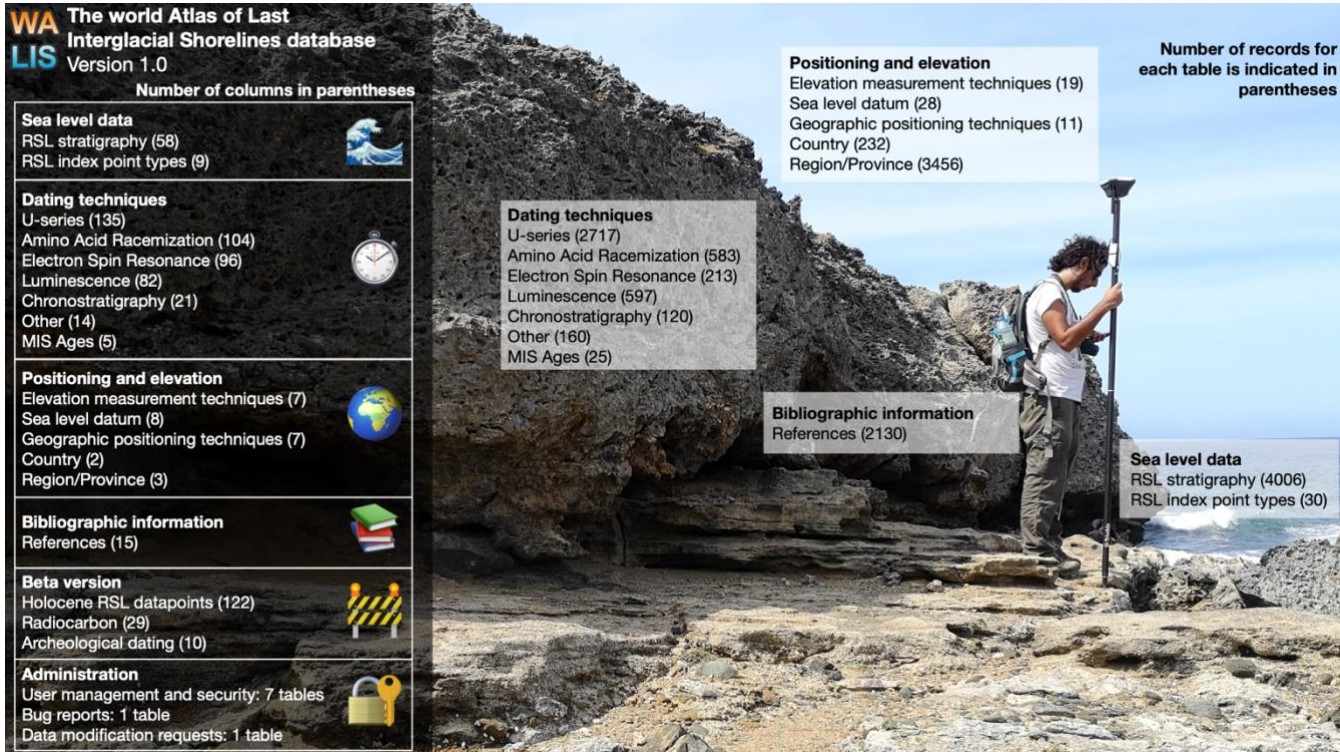

**Figure 2 Summary of the number of columns in each table and the amount of data per table in WALIS 1.0. Photo by A. Rovere (2019). Location: Boca Cortalein, Curaçao, Netherlands Antilles. In the photo: geologist Ciro Cerrone measuring the elevation of a Pleistocene beach deposit using Real Time Kinematics GNSS.**

WALIS has a complex structure, with a series of one-to-many and many-to-many links between tables (Figure 3). One example of a one-to-many relationship is the one between the tables "RSL stratigraphy" (containing the stratigraphic descriptions of sea-level index points) and "Elevation measurement techniques" (including details on the techniques used to measure site or sample elevation). Each record in the "Elevation measurement technique" table can be associated with many "RSL stratigraphy" data points, as the same measuring technique can be applied to many sites. But each "RSL stratigraphy"

record can be associated with only one "Elevation measurement technique". The relationship between the "RSL stratigraphy" and any table containing dated samples (e.g., amino acid racemization, electron spin resonance, and U-series) is instead an example of a many-to-many relationship. Multiple dated samples can characterize one "RSL stratigraphy" data point (e.g., replicates of the same dated material or different samples within the same stratigraphic layer). But it is also true that a single dated sample can be used within various "RSL stratigraphy" records. This can happen when different sites

containing sea-level index points correlate to the same sample dated, for example, with U-series. The existence, in WALIS, of many-to-many relationships between sea-level proxies and dated samples is a crucial difference with both the Holocene sea-level database (Khan et al., 2019) and previous Last Interglacial sea-level databases (Hibbert et al., 2016; Pedoja et al.,



2014; Ferranti et al., 2006). In these databases, the relationship between these two entities is always one-to-one (e.g., one radiocarbon sample corresponding to one peat layer indicating a former RSL).


The links between database tables are shown in Figure 3. One key aspect of WALIS is that the links are not defined in MySQL but are managed in the PHP interface. Taking the examples of the one-to-many and many-to-many relationships described above, in the "RSL stratigraphy" there is an "Elevation measurement technique" type integer field, which is filled via the PHP interface with the corresponding ID in the "Elevation measurement techniques" table. Similarly, in the "RSL

stratigraphy" table, there is a "U-series" type varchar field that, upon selection in the PHP interface, is filled with the corresponding ID(s) in the "U-series" table as comma-separated values. Unique constraints and orphans are also managed via the PHP interface, and all relationships are coded into the python scripts used to extract the data (Figure 1). While we recognize that this makes WALIS an "unorthodox" database from the SQL standpoint, we found that this is the most effective way to manage the inter-table relationships' complexity without affecting the interface's usability.





Figure 3 Database structure for WALIS. Note that the relationships among tables are managed via the PHP interface and are therefore not included in the database. Also not included in this scheme are the tables to report Holocene data, which are still in beta testing version.

**3.2 The WALIS interface (version 1.8.0)**

The interface to the MySQL database was built in PHP 7.3 using Scriptcase, a Rapid Application Development software. The main components of the interface (current version: 1.8.0) are shown in Figure 4. On the left side, a menu allows navigating the different pages of the application. Windows for data insertion can be browsed in a tab space in the central part of the application. Below the tab space is the main window, where the active tab is shown. The data insertion is done inside the active tabs in the main window, where a data compiler can find the fields to fill and help tips to guide in the compilation

of the database.





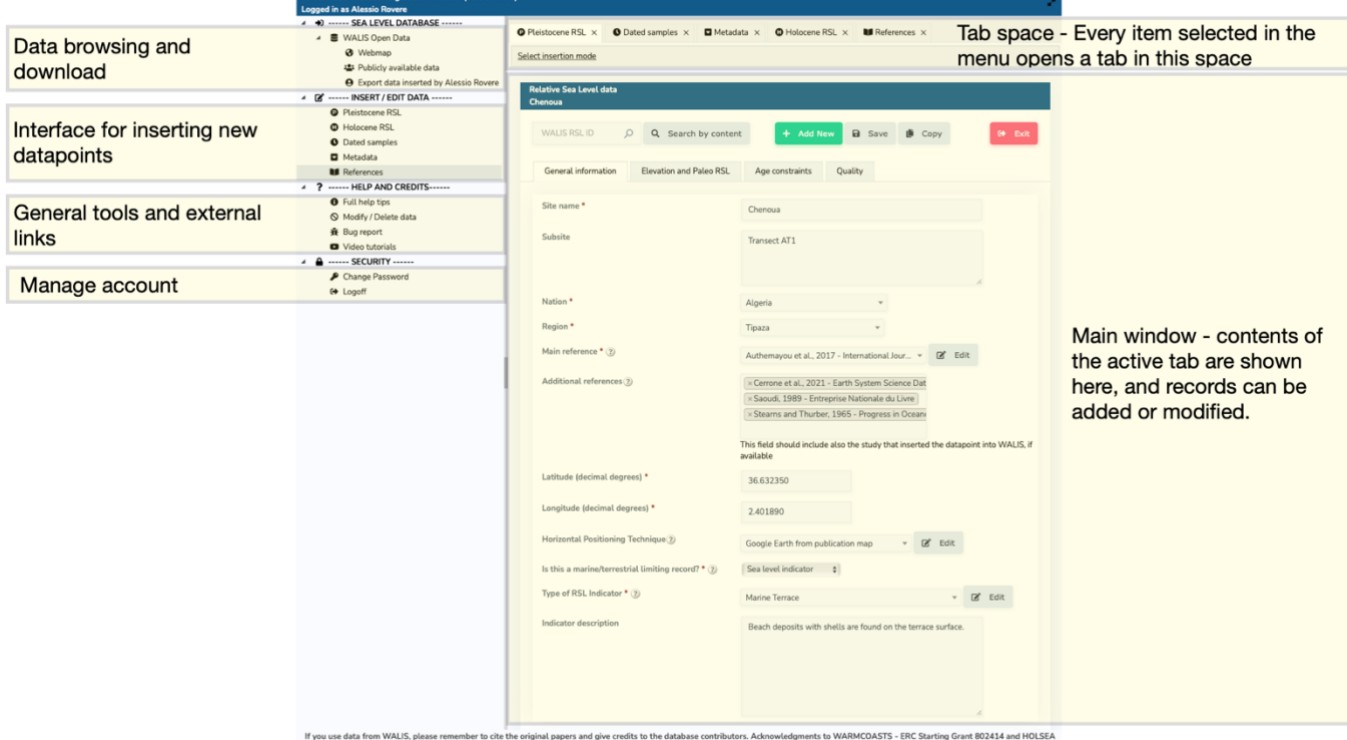

**Figure 4 Annotated screenshot of the WALIS database interface, version 1.8.0.**

### 3.3 Data visualization (version 1.0)

To ensure rapid access to the data in WALIS, we also prepared a map interface that allows end-users to browse the contents of the database (Garzón and Rovere, 2021). This visualization interface is built with the R package Shiny, and it uses a simplified version of the database, exported via a python script (Table 1). The interface's main page allows an end-user to select data from different filters or geographic queries. The chosen data on the first page can be browsed and downloaded in CSV format within the second page of the interface (Figure 5).



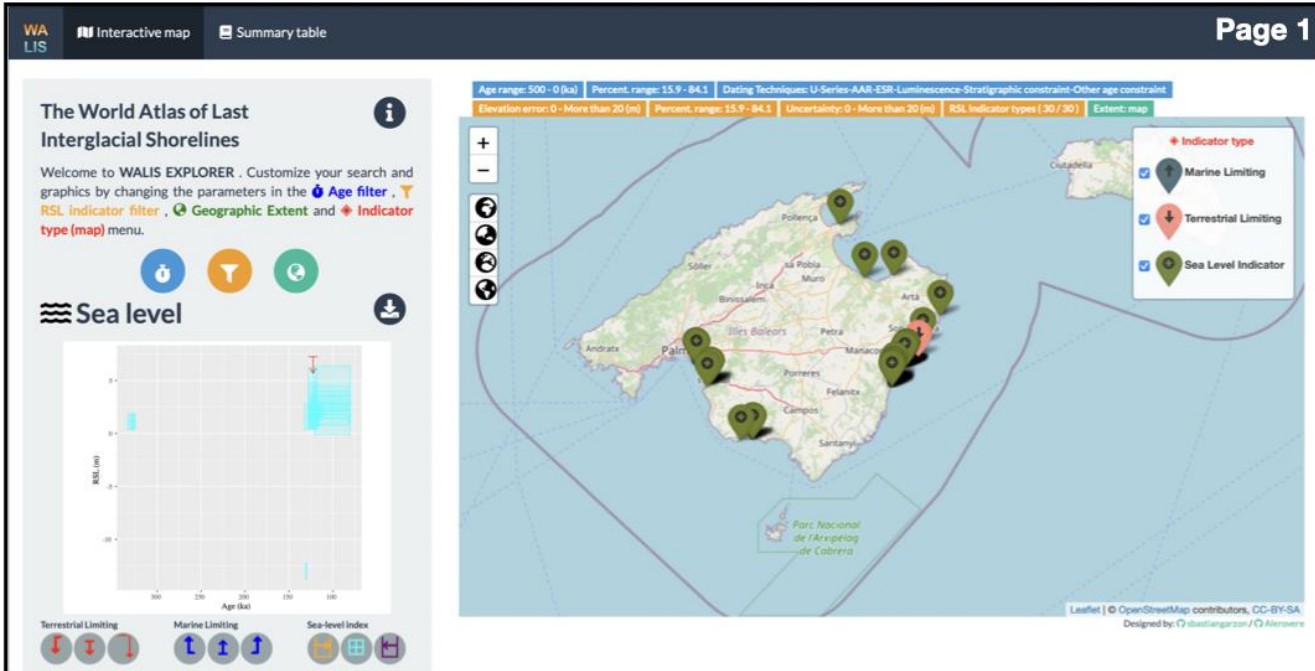

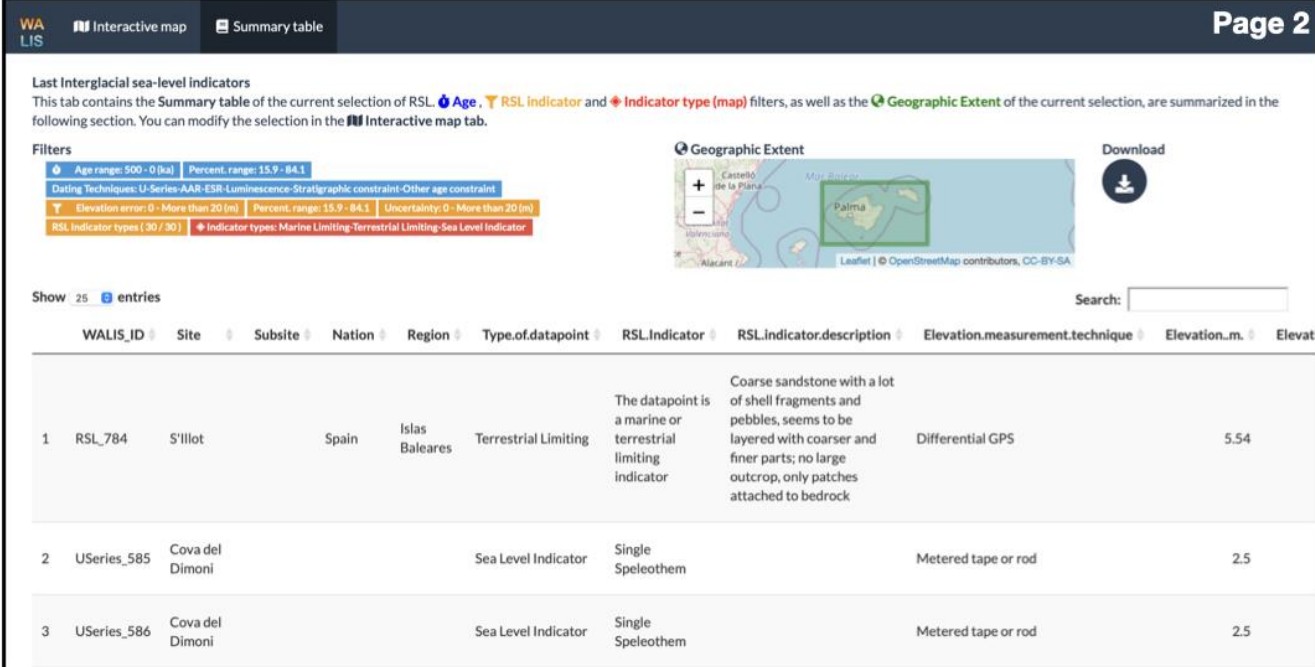


**Figure 5 Screenshots of the two pages of the WALIS data visualization, version 1.0.**



## 3.4 Query and plotting scripts

With the release of WALIS 1.0, we prepared a repository that includes several python scripts that can be executed via
Jupyter Notebooks. The scripts allow querying WALIS by selecting data compilers by name or geographic boundaries.
Then, they connect records via one-to-one and many-to-many links, and associate database column names with formatted
labels. After these scripts are run, it is possible to export the queried data in different formats (CSV, XLSX, GeoJson) or
make data exploratory plots, such as maps or histograms for various fields (Table 1).

## 3.5 Documentation

The documentation associated with WALIS is maintained in GitHub and served via a ReadTheDocs webpage built with
Sphinx (Brandl, 2021). The documentation contains details on the main tables of WALIS, and a description for each field of
the database, with guidelines on which values or details are expected to be included by the data compiler. Via the
ReadTheDocs page, the documentation may be searched and exported as a PDF file.

**Table 1 Main resources associated with WALIS, with repositories on GitHub and direct link to the application (if available)**

| Citation | Description | GitHub page | Direct link |
|---|---|---|---|
| | PHP database interface | N/A | https://warmcoasts.eu/world-atlas.html |
| Rovere et al., 2022 | Repository with python scripts to perform queries, download data, or make common plots. It also contains the full database in different formats | https://alerovere.github.io/WALIS/ | |
| Rovere et al., 2020 | Description of the database tables and fields, with help for the compilation of WALIS | https://github.com/Alerovere/WALIS_Help.git | https://walis-help.readthedocs.io/ |
| Garzón and Rovere, 2021 | R code for the visualization interface | https://github.com/Alerovere/WALIS_Visualization | https://warmcoasts.shinyapps.io/WALIS_Visualization/ |

## 4 Database contents

The data included in WALIS 1.0 were compiled primarily within the ESSD Special Issue (Table 2). Some of the papers in
the Special Issue have global scope, such as those collecting data for U-series on corals, speleothems, and MIS 5c/5a
(respectively: Chutcharavan and Dutton, 2021a; Dumitru et al., 2021; Thompson and Creveling, 2021b). Most manuscripts
are instead focused on specific regions. Two manuscripts for which data is available in WALIS 1.0 are only available as
preprints. At the same time, data from one paper published outside the Special Issue were submitted by the lead author to
WALIS (Steidle et al., 2021). In one case (not shown in Table 2), the WALIS U-series structure was used to report on Last
Interglacial coral and speleothem ages in the Mediterranean Sea, adding more samples to those currently in WALIS for that



region (Pasquetti et al., 2021). These data were not compiled via the interface; therefore, they will be included in the subsequent versions of WALIS.

**Table 2 List of manuscripts describing data compiled in WALIS format and associated datasets. * Preprints, not peer-reviewed; ** Data from this paper are available within Version 1.0 of the global database but are not included in the ESSD Special Issue and not uploaded in Zenodo as a regional dataset.**

| Area | Manuscript | Dataset |
|---|---|---|
| Global (U-Series on corals) | Chutcharavan and Dutton, 2021a | Chutcharavan and Dutton, 2021b |
| Global (U-Series on speleothems) | Dumitru et al., 2021 | Dumitru et al., 2020 |
| Global (MIS 5c and MIS 5a) | Thompson and Creveling, 2021b | Thompson and Creveling, 2021a |
| Bahamas, Florida, Turks and Caicos | Dutton et al., 2021 | Dutton et al., 2021 |
| Gulf of Mexico | Simms, 2021 | Simms, 2020 |
| Southwestern Atlantic | Rubio-Sandoval et al., 2021a | Rubio-Sandoval et al., 2021b |
| Pacific Coast of North America | Muhs, 2022 | Muhs et al., 2021 |
| Southeast South America | Gowan et al., 2021 | Gowan et al., 2020 |
| Pacific coast of South America | Freisleben et al., 2021 | Freisleben et al., 2020 |
| Japan | Tam and Yokoyama, 2021 | Tam and Yokoyama, 2020 |
| Southeast Asia | Maxwell et al., 2021a | Maxwell et al., 2021b |
| Korean peninsula | Ryang et al., 2022 | Ryang and Simms, 2021 |
| Tropical Pacific Islands | Hallmann et al., 2021 | Hallmann and Camoin, 2020 |
| New Zealand | Ryan et al., 2021 | Ryan et al., 2020 |
| East Africa and West Indian Ocean | Boyden et al., 2021b | Boyden et al., 2021a |
| Southern Africa | Cooper and Green, 2021 | Cooper and Green, 2020 |
| West Mediterranean | Cerrone et al., 2021b | Cerrone et al., 2021a |
| Glaciated Northern Hemisphere | Dalton et al., 2022 | Dalton et al., 2021 |
| Northwest Europe | Cohen et al., 2021a | Cohen et al., 2021 |
| Eastern Mediterranean* | Mauz and Elmejdoub, 2021; Mauz et al., 2020 | Mauz, 2020; Sivan and Galili, 2020; Zomeni, 2021 |
| Yucatan (Mexico) cave deposits** | Steidle et al., 2021 | Included in WALIS 1.0 with no regional datasets published |


## 4.1 Sea level and age data points

WALIS 1.0 includes 4545 sea-level proxies (Figure 6), with data and metadata extracted from 2130 references spanning more than one century of published scientific literature. The sea-level proxies in WALIS 1.0 consist of 3311 sea-level index points from coastal sites (containing, for example, fossil beach deposits or marine terraces), for which it is possible to

identify a relationship with the paleo sea level via the "indicative meaning" (Shennan et al., 2015). Other points indicate that paleo sea level was above ("marine limiting" points, $n$=285) or below ("terrestrial limiting points, $n$=410) the measured stratigraphy. Examples of marine limiting points are fossil subtidal sands, while terrestrial limiting points are, for example, fossil dunes. The database also contains 463 sea-level index points from fossil corals (most of them compiled by Chutcharavan and Dutton, 2021a), that can be correlated to paleo sea level when enough information on paleowater depth is

given. In WALIS are also included 76 phreatic overgrowths on speleothems, which are a particular morphotype of cave deposit having formed in association with paleo sea level (Dumitru et al., 2021).

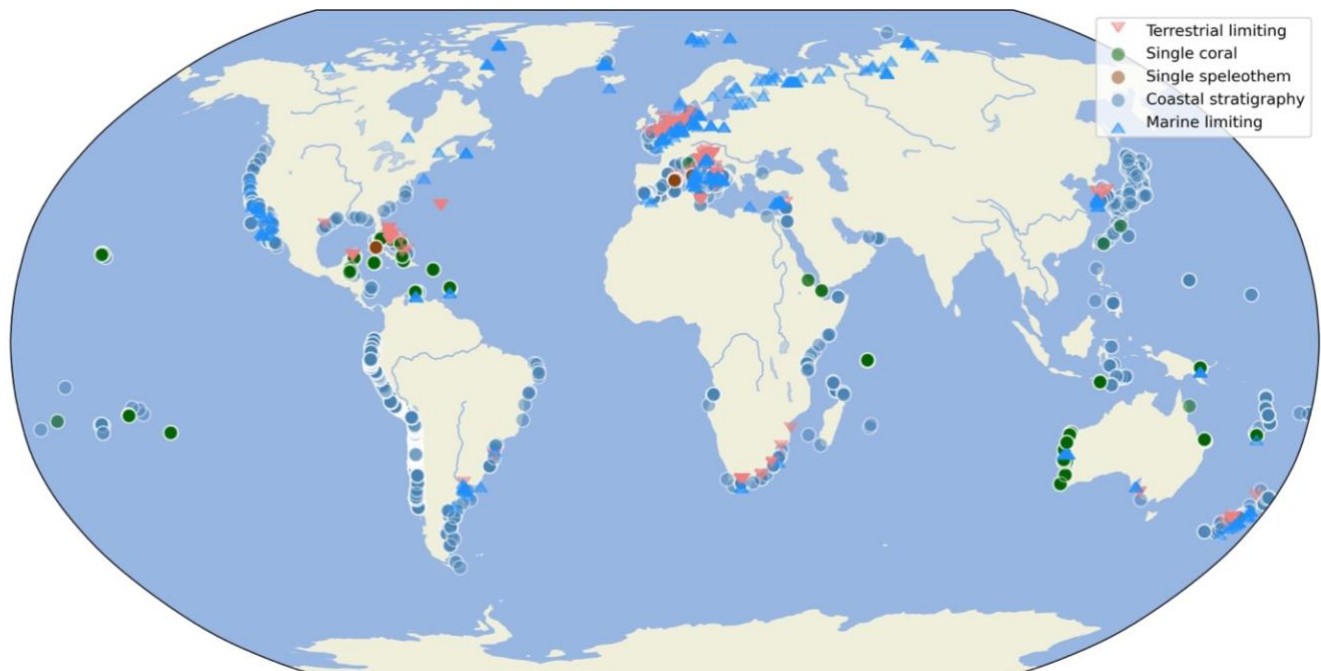

**Figure 6 Location of sea-level proxies included in WALIS 1.0, divided by category. The code used to produce this image is**
**available from Rovere et al., 2022. Map made with Natural Earth, background from https://www.naturalearthdata.com. The code**
**used to produce this image is available from Rovere et al., 2022.**

Each sea-level index point is associated with one or more dated samples or with non-radiometric chronological constraints,

such as, for example, biostratigraphic markers. In total, WALIS 1.0 includes 4110 dated samples. These include corals, cave

deposits, mollusks, or oolites dated via U-series ($n$=2717). Upon selecting the material dated by U-Series techniques, the

interface requests the compiler to fill in different fields specific to the material chosen. WALIS 1.0 also contains a relevant

number of samples dated with luminescence ($n$=597), amino acid racemization ($n$=583), and electron spin resonance

($n$=213). Additional chronological constraints include 120 chronostratigraphic or biostratigraphic records (e.g., pollen zones)

and 160 age determinations of "other" type. Examples of the latter are tephra layers or finite radiocarbon ages, or ages

inferred to be beyond the range of the radiocarbon method. Each sea-level proxy must be correlated with at least one

chronological constraint. There is no upper limit on the number of constraints associated with a single sea-level proxy. In the

interface, it is also possible to select minimum or maximum ages, indicating if a sea-level proxy is older or younger than a

given sample. Together with the definition of a sea-level proxy as a sea-level index point, marine or terrestrial limiting, this

gives rise to nine possible combinations of paleo sea level and age information associated with a sample in WALIS (Figure

7).

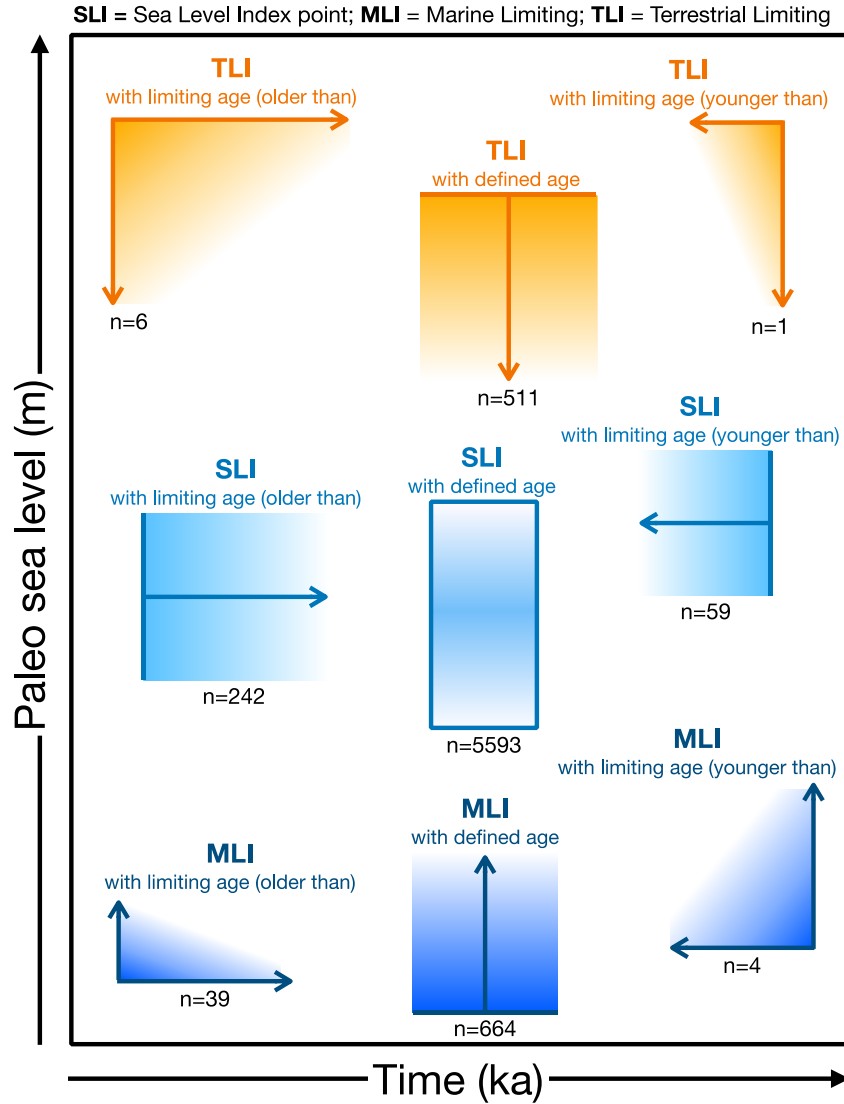


**Figure 7 Nine possible combinations of sea-level proxies and age determination types in WALIS 1.0. The number on the bottom of each type indicates its frequency within the database.**

**4.2 Types of sea-level proxies**

In WALIS 1.0, paleo sea levels are interpreted from thirty different sea-level indicators. In Figure 8, we group them into four

general categories. In the current version, geomorphological sea-level indicators (such as marine terraces, shoreline angles or tidal notches) are the most represented category. The high number of records for this category is driven by the data in Freisleben et al. (2020), who used a tool called TerraceM (Jara-Muñoz et al., 2016) to systematically map the inner margin

of marine terraces along the Pacific coasts of South America. These authors report 1953 single points, representing more

than 80% of the marine terrace data points in WALIS (and 43% of the datapoints in WALIS 1.0). Biological indicators (i.e.,

single corals or coral reef terraces) and limiting points (marine or terrestrial) contain relatively fewer records. The lowest

occurrence in the database are indicators of depositional origin. This category includes several proxies, ranging from widely-

defined fossil beach deposits and beachrocks (Mauz et al., 2015), to beach ridges (Otvos, 2000) or lagoonal deposits.

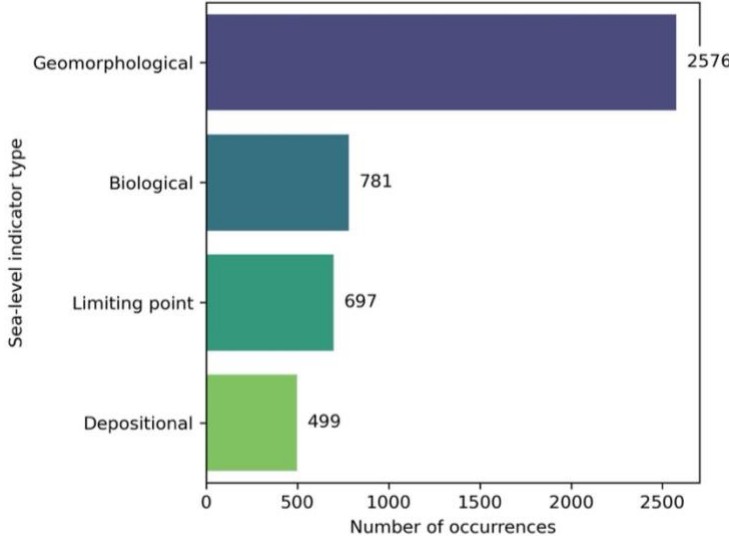

**Figure 8 Percentage of occurrence in WALIS of main categories of sea-level proxies. The code used to produce this image is available from Rovere et al., 2022.**

### 4.3 Elevation metadata

Data and metadata on elevation measurement and the vertical datum used are essential to describe a sea-level proxy, as they

affect the uncertainties associated to paleo RSL. For this reason, WALIS includes two tables containing information on

elevation measurement methods and sea-level datums used to report the elevation of samples and sea-level proxies in the

field. Each sea-level proxy and each sample in WALIS 1.0 must relate to one record in these two tables. Looking at broad

categories of elevation measurement techniques (Figure 9), it is evident that only a fraction of sea-level proxies in WALIS

1.0 has been measured with techniques allowing to survey elevations with sub-metre (or better) accuracy (e.g., differential

GNSS or total station). Elevations for almost half of the sea-level proxies in WALIS 1.0 were gathered from digital elevation

models or topographic maps. This high percentage is driven by the 1953 marine terraces measured on digital elevation

models by Freisleben et al. (2021). Also, elevation measurement methods are often labelled in WALIS as "not reported",

meaning that there was insufficient information in the literature to discern how the elevation of a sea-level proxy was

initially measured.





Looking at broad categories of sea-level datums to which elevation measurements have been attributed (Figure 9), there is a large number of data referred to either ordnance or geodetic datums. The high frequency of the latter is driven by the large number of terraces reported by Freisleben et al., 2021, who referred their measurements to the global EGM08 geoid. If this source is excluded, the vertical datum for most records in WALIS is either not reported or referred generally to "mean sea level" with no further specifications. Only a small fraction of data is referred to a tidal datum, and very few datapoints were

referred to a biological datum (e.g., the height of living coral microatolls).

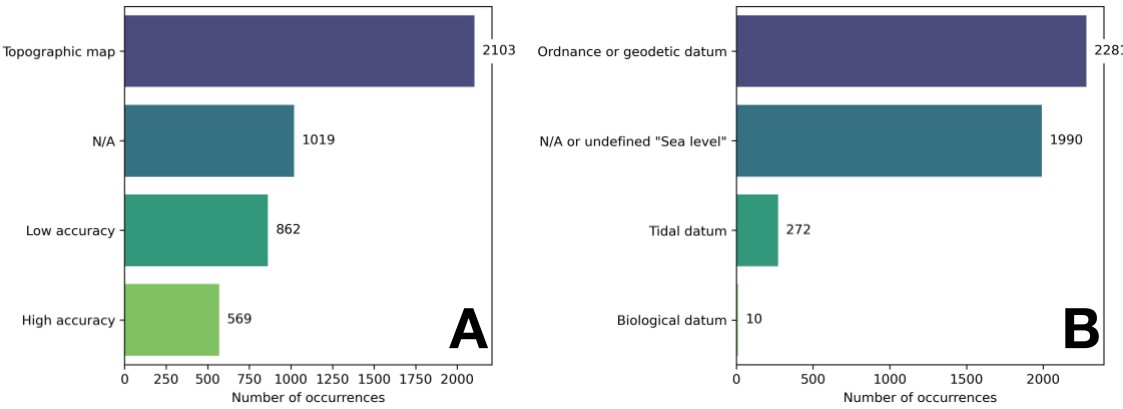

**Figure 9 Elevation measurement methods (A) and sea level datums (B) associated with sea-level proxies in WALIS. The code used to produce this image is available from Rovere et al., 2022.**

**4.4 Relative sea-level estimates from WALIS data**

As shown above in Figure 8, sea-level proxies in WALIS 1.0 can be divided into four broad categories. However, there are differences in how relative sea-level (RSL) information is inserted in the WALIS interface for three common types of index points: from stratigraphy (including geomorphological proxies), speleothems, or corals (Figure 10). For any sea-level index point, regardless of the type, the WALIS interface requests as mandatory information the elevation of the proxy and the associated 2σ error, which should also include datum uncertainty.


Regarding RSL index points from stratigraphy, the WALIS interface calculates paleo RSL from the upper and lower limits of occurrence of the proxy in modern analogues, which are requested as mandatory fields. Using the concept of indicative meaning (Shennan, 1982; Shennan et al., 2015) and associated formulas (Rovere et al., 2016), the interface calculates paleo RSL and 2-sigma uncertainties that are then saved in the database. In the case of sea-level index points from speleothems, in

WALIS 1.0 represented by Phreatic Overgrowth on Speleothems (POS), the compiler is required to insert the values of paleo RSL and associated 2-sigma uncertainty. In the case of sea-level index points from single corals, the WALIS interface requests the insertion of upper, lower, and modal limits of occurrence of living specimens. For corals, the interface does not calculate paleo RSL. However, in the code used to prepare data for the ShinyApp interface (Rovere et al., 2022), we include



scripts to compute a gamma distribution from the upper and lower limits of occurrence of corals used as index points (shown

in Figure 10), an approach similar to that proposed by Hibbert et al., 2016.

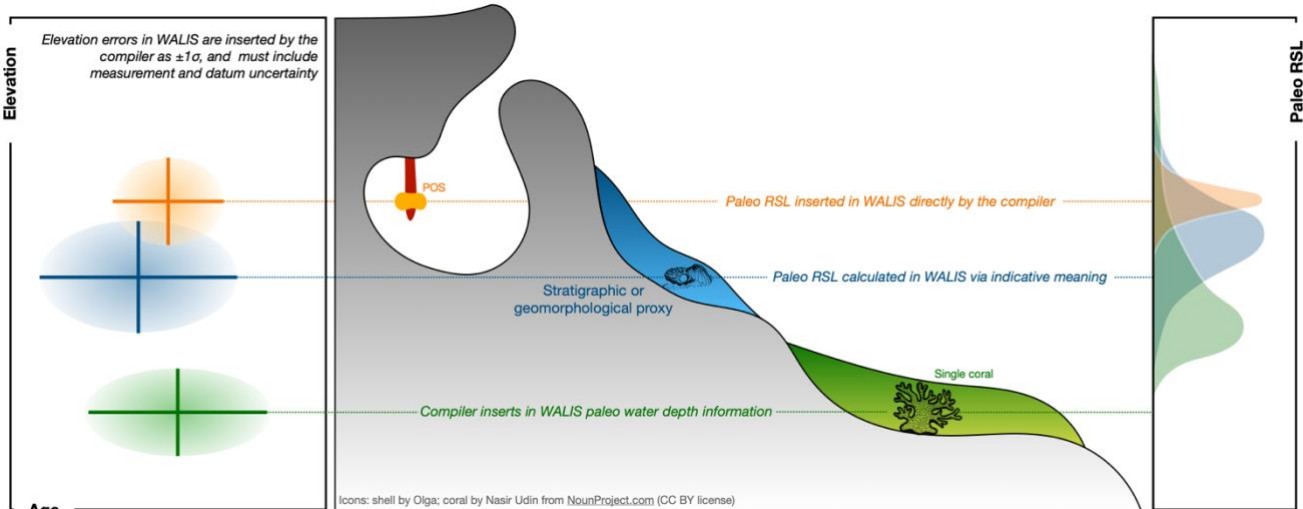

**Figure 10 Three types of index points in WALIS 1.0 and associated elevation (left) and paleo RSL information (right). The figure represents the ideal case of three index points formed at the same time (overlapping ages within error bars), surveyed at different**
**elevations, but resulting in a coherent paleo RSL history (overlapping paleo RSL) once interpretation of proxies is included. POS= Phreatic Overgrowth on Speleothems.**

## 5 Principal limitations of WALIS 1.0

An essential aspect of any database is the assessment of limitations in its structure and in the data it contains. The WALIS structure was agreed upon by a pool of scientists who are experts on different regions and dating techniques. It was also
modified as needed during the compilation of version 1.0, following the suggestions of data compilers. Within the interface, there is the possibility to report "bugs" or recommendations to the administrators, helping future developments of the WALIS interface and database structure. The data included in WALIS 1.0 underwent peer-review together with the associated papers, and most data included in WALIS 1.0 has been subject to peer-review via their original publication. However, errors or inaccuracies may be present in WALIS 1.0. For this reason, the interface includes the possibility for both
data compilers and end-users to indicate issues with specific data points. Correcting the problems identified in this way will likely lead to future WALIS versions, allowing the tracking of changes and sharing credit with new contributors.

### 5.1 Data quality

In several tables within WALIS, there are fields to help end-users understand the quality of the data. We remark that, in this context, quality does not necessarily refer to accuracy or precision, but is a subjective measure that can reflect, for example,
more information associated with the data point that leads it to be considered more robust.  In the "RSL from stratigraphy"





table, data compilers were asked to score, on a scale from 0 to 5, the age and RSL information quality of each datapoint inserted. A general guide on how to score the records was given in Rovere et al., 2020 (Table 3). However, its use was not strictly enforced within the WALIS Special Issue, and although the guidelines are designed to facilitate objective analysis of the data, we recognize there is still potential for subjective interpretation.


For studies following this suggested ranking scale, comparing quality scores among areas is possible. Examples are the records of Argentina (Gowan et al., 2020) and Brazil (Rubio-Sandoval et al., 2021a), shown in Figure 11. Comparing the scores in these two datasets shows that, on average, records in Argentina have slightly higher age quality than those in Brazil. However, the RSL information for some sites in Brazil reaches the "excellent" score, which is not achieved by any

record in Argentina. Ideally, those sites might be targeted to gauge whether it may be possible to improve their age control.

**Table 3 Quality scores as suggested by the WALIS guidelines (verbatim from Rovere et al., 2020, originally published under the CC-BY 2.0 license). Scores vary from 0 (Rejected) to 5 (Excellent).**

| Quality of RSL data | Score | Quality of age constraints |
|---|---|---|
| Elevation precisely measured, referred to a clear datum and RSL indicator with a very narrow indicative range. Final RSL uncertainty is submetric. | 5 | Very narrow age range, e.g. few ka, that allow the attribution to a specific timing within a substage of MIS 5 (e.g., 117±2 ka) |
| Elevation precisely measured, referred to a clear datum and RSL indicator with a narrow indicative range. Final RSL uncertainty is between one and two meters. | 4 | Narrow age range, allowing the attribution to a specific substage of MIS 5 (e.g., MIS 5e) |
| Uncertainties in elevation, datum or indicative range sum up to a value between two and three meters. | 3 | The RSL data point can be attributed only to a generic interglacial (e.g., MIS 5) |
| Final paleo RSL uncertainty is higher than three meters | 2 | Only partial information or minimum age constraints are available |
| Elevation and / or indicative range must be regarded as very uncertain due to poor measurement / description / RSL indicator quality | 1 | Different age constraints point to different interglacials |
| There is not enough information to accept the record as a valid RSL indicator (e.g., marine or terrestial limiting) | 0 | Not enough information to attribute the RSL data point to any pleistocene interglacial. |





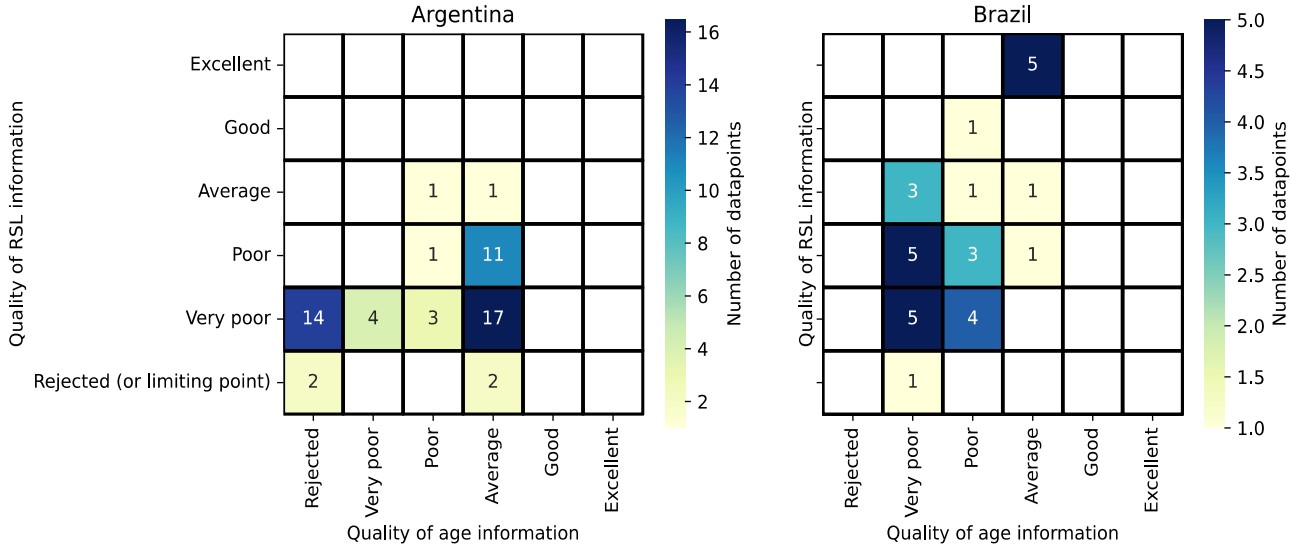

**Figure 11 Heatmap summarizing the quality of age and RSL information for data in Argentina (Gowan et al., 2020) and Brazil (Rubio-Sandoval et al., 2021a). Labels inside each cell detail the number of sites with the corresponding RSL/age quality scores. The color of each cell is related to the relative frequency of each duplet of scores. The code used to produce this image is available from Rovere et al., 2022.**

Samples dated with radiometric techniques have a mandatory field where data compilers indicate whether the data point is "accepted" or "rejected". The rationale of this field is to allow the data compiler to insert samples rejected by original authors and retain unsuccessful dating attempts for future reference. Such fields were expanded in the U-series ages on corals template thanks to the input by Chutcharavan and Dutton, 2021a. These authors proposed inserting fields in the database to report different screening protocols applied to existing ages. Investigating whether U-series approach on corals might also be used for other radiometric dating techniques represents potential ground for future WALIS versions.

**5.2 Geographic gaps**

In compiling data for WALIS 1.0, the aim was to include as many sites as possible globally, including all relevant information related to sea-level proxies and associated dating methods. While we achieved both goals, comparing the data included in WALIS 1.0 with the sites reported in the extensive review by Pedoja et al., 2014, it is possible to highlight areas where Last Interglacial sea-level proxies might be present but are not available in the WALIS standard format. These areas are shown in Figure 12 and are discussed below, together with relevant or most recent works, which may be used as starting point for the inclusion of these areas in the subsequent versions of WALIS.

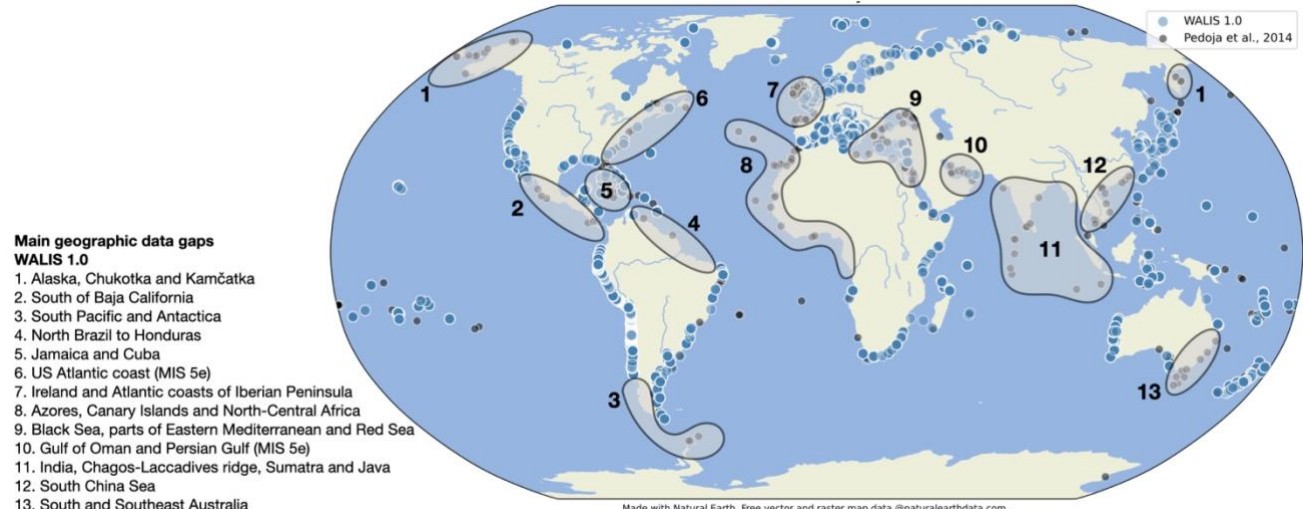

**Main geographic data gaps**
**WALIS 1.0**
1. Alaska, Chukotka and Kamčatka
2. South of Baja California
3. South Pacific and Antactica
4. North Brazil to Honduras
5. Jamaica and Cuba
6. US Atlantic coast (MIS 5e)
7. Ireland and Atlantic coasts of Iberian Peninsula
8. Azores, Canary Islands and North-Central Africa
9. Black Sea, parts of Eastern Mediterranean and Red Sea
10. Gulf of Oman and Persian Gulf (MIS 5e)
11. India, Chagos-Laccadives ridge, Sumatra and Java
12. South China Sea
13. South and Southeast Australia

**Figure 12 Comparison between sea-level proxies in WALIS 1.0 and those in the global review by Pedoja et al. (2014), indicating areas where geographic data gaps are present (see details in the main text). Map made with Natural Earth, background from https://www.naturalearthdata.com.**

Starting from the northernmost part of the North American continent, areas not included in WALIS 1.0 include Last Interglacial shorelines reported in Alaska, along the coasts of the Lisburne and Seward peninsulas, the latter including the Nome coastal plain where sea-level records left by several sea-level highstands have been reported (Brigham-Grette and Hopkins, 1995; Goodfriend et al., 1996). On the Russian side of the Bering Strait, in the Chukotka Peninsula, the Last Interglacial sequences are reported by Khim et al. (2001) and Brigham-Grette et al. (2001). South of this area, Quaternary marine and coastal sequences were also reported on the Eastern side of the Kamčatka peninsula (Pedoja et al., 2013).

Another data gap in WALIS 1.0 is evident on the Pacific Coasts of North and Central America. Potential sites in this area (including the Mexican states south of Sonora and the Pacific coast of Central America) are discussed in Muhs (2022), who highlights that some studies on these locations are present but do not contain enough details to be inserted in WALIS. We also emphasize that WALIS 1.0 does not include the data reported by a recent paper detailing the marine terraces of Santa Cruz Island, California, which appeared in the literature during the compilation of the Atlas (Muhs et al., 2021b).

There are virtually no Last Interglacial shorelines along the South American Pacific coasts extending south of 75° S latitude. The absence of such features in this vast area is probably due to the presence of the Patagonian ice sheet during the Last Glacial Maximum, which has likely eroded most, if not all, Last Interglacial coastal sections in this area. Going southwards, surfaces of marine origin, likely of Pleistocene age, are reported on the South Shetland Islands (Navas et al., 2006; López-Martínez et al., 2016) but not included in WALIS 1.0. It is unclear whether the chronological constraints on those landforms are robust enough to be inserted in WALIS.



On the western coasts of Central Atlantic, another vast area with a substantial lack of records in WALIS 1.0 includes the shores of Northern Brazil, French Guyana, Suriname, Guyana, Venezuela, and the Caribbean Sea coasts of Colombia,
Panama, Costa Rica, Nicaragua, and Honduras. Potential sites in this vast area are discussed in Rubio-Sandoval et al. (2021a). However, studies with enough metadata are missing, except offshore Colombia, on the San Andrés and Providencia Islands. Regarding the Caribbean Sea, WALIS 1.0 includes the most relevant data from this region. However, additional sites that may be inserted in WALIS are present in Jamaica (Mitchell et al., 2001, 2006) and Cuba (Muhs et al., 2017; Schielein et al., 2020).


For the US Atlantic coast (excluding the Florida Keys), the only data included in WALIS 1.0 are those related to MIS 5a/5c (Thompson and Creveling, 2021b). Former studies suggested that these stages attained similar elevations to MIS 5e in this area due to glacial isostatic adjustment processes (Wehmiller et al., 2004) therefore, outcrops of this age might have been eroded by later highstands. However, there are reports of MIS 5e successions scattered along the coast (Wehmiller et al.,
2010, 2012; O'Neal and McGeary, 2002; Wright et al., 2009), which should be screened for insertion in WALIS. Also, as already mentioned in the previous sections, the Last Interglacial amino acid racemization data obtained from samples of the US Atlantic coast (Wehmiller et al., 2021) should be included in the future versions of WALIS. In Bermuda, data from corals (Chutcharavan and Dutton, 2021a) and cave deposits (Dumitru et al., 2021) are included in WALIS 1.0. Still, more sites and dated samples are reported in a recent review by Muhs et al. (2020). Their inclusion in the subsequent versions of
WALIS must be considered a priority, given the importance of this area for unraveling land motion changes driven by glacial isostatic adjustment in the North Atlantic.

On the East Atlantic, Pedoja et al., 2014 report that Last Interglacial sea-level proxies were mapped in Ireland by Orme ( 1966). While it seems unlikely that this study contains enough metadata to be considered for WALIS, some of the
Quaternary deposits presented in this early study might have been the subject of successive dating (e.g., Gallagher and Thorp, 1997), thereby meeting the standard for their inclusion in subsequent versions of WALIS. Last Interglacial data missing from WALIS 1.0 may also be present on the Atlantic coasts of the Iberian Peninsula (Alonso and Pagés, 2007; Meireles and Texier, 2000; Benedetti et al., 2009). In the Azores islands, Last Interglacial deposits have also been reported (Ávila et al., 2009, 2015) but are not included in WALIS 1.0. Only U-series ages with no indication on paleo sea level are
reported in WALIS 1.0 from the Canary and Cape Verde archipelagos. However, enough metadata might be present in some papers (Zazo et al., 2010, 2007; Muhs et al., 2014) to upgrade these points to sea-level proxies in the subsequent versions of WALIS.



Within the Mediterranean Sea, data from several studies was inserted in WALIS 1.0. However, some areas lack records,
mainly in the Eastern part of the basin. There seem to be little to no studies on the coastal Quaternary of Libya, except for
one site near Alexandria inserted in WALIS 1.0 (Mauz and Elmejdoub, 2021). While sites in Israel and Cyprus have been
inserted in WALIS 1.0 (Sivan and Galili, 2020; Zomeni, 2021), there are no sites in WALIS for Lebanon, Syria, Turkey, and
Greece. In these areas, Last Interglacial sea-level proxies are reported in the literature (e.g., Sanlaville, 1974; Dodonov et al.,
2008; Tarı et al., 2018; Gaki-Papanastassiou et al., 2009) and should be screened to gauge whether enough metadata are
available for their insertion in the following versions of WALIS. Also, Last Interglacial sea-level proxies from the Black Sea
are reported by Pedoja et al. (2014), but are not yet present in WALIS 1.0. The published records in these areas need to be
evaluated and eventually included in the subsequent versions of WALIS.

There is a substantial lack of data in WALIS 1.0 along the Atlantic coasts of Northern and Central Africa. However, there
are several recent studies detailing Last Interglacial marine and coastal sequences in Morocco (Barton et al., 2009; Plaziat et
al., 2008, 2006; Rhodes et al., 2006), Mauritania, and Senegal (Giresse et al., 2000), where data might be gathered for
subsequent WALIS versions. It appears that there are only a few other reports of Last Interglacial shorelines (e.g., Gregory,
1962) south of Senegal, the first southwards being those in Angola inserted in WALIS 1.0 by Cooper and Green (2021).

Another area where more data could be added to WALIS is the Red Sea. Here, several U-series ages on corals have been
reviewed by Chutcharavan and Dutton (2021a), some of which were identified as sea-level index points from corals. Further
sea-level index points might be retrieved from studies reporting on Last Interglacial shorelines in Saudi Arabia (Dullo, 1990)
and the Gulf of Suez (Parker et al., 2012; Bosworth and Taviani, 1996). On the other side of the Arabian peninsula, in the
Gulf of Oman and the Persian Gulf, the only data inserted in WALIS 1.0 are those related to MIS 5a/5c proxies (Thompson
and Creveling, 2021b). However, potential MIS 5e sea-level proxies are reported in the literature from Oman (Falkenroth et
al., 2020, 2019), United Arab Emirates, Qatar (Williams and Walkden, 2002), and Iran (Oveisi et al., 2007; Pirazzoli et al.,
2004). These studies must be screened to gauge whether valid sea-level index points can be inserted in the subsequent
versions of WALIS.

There are no data in WALIS 1.0 for India and the Chagos-Laccadive ridge. Key references from which data might be
retrieved are Bhatt and Bhonde (2006), Banerjee (2000) (for India), Woodroffe (2005), Gischler et al. (2008) (for the
Chagos-Laccadive Ridge). We could not find publications reporting data for Last Interglacial successions in Thailand,
Malaysia, and the islands of Sumatra and Java. WALIS 1.0 is also missing data from the South China sea, albeit some
potentially Last Interglacial sites were reported in Pedoja et al., 2008.




In Australia, the Last Interglacial data included in WALIS 1.0 are mainly derived from corals (Chutcharavan and Dutton, 2021a) along the Western coast and the Great Barrier Reef. Thompson and Creveling (2021b) report MIS 5a/5c in South Australia. Further data on MIS 5e shorelines from strand plains in South Australia are available from several studies (Murray-Wallace and Belperio, 1991; Murray-Wallace, 2002; Murray-Wallace et al., 2016; Pan et al., 2018), and should be included in the subsequent versions of WALIS.

### 5.3 Future developments

There are four main ways in which WALIS could be improved in the future, detailed below. More will likely emerge as the database starts to be used by end-users.

1) As mentioned above, the database structure is, at the moment, a collection of tables with no links defined in MySQL. Creating such connection would imply a partial restructuring of the interface. However, this would improve the overall structure of the WALIS database.

2) One of the main aims of WALIS is to standardize the reporting of sea-level proxies and dated samples for the Last Interglacial. For reconstruction of former sea levels from WALIS data, the focus is on relative sea-level proxies. The paleo RSL calculated from these proxies (Figure 10) represents the sum of global mean sea-level changes (driven by ice melting and thermal expansion) and local effects due to land motions caused by subsidence, tectonic uplift, glacial isostatic adjustment, and other processes. Fields to report on rates of uplift or subsidence are present in WALIS, with the caveat that such estimates must be independent from the Last Interglacial sea-level record itself. Uplift or subsidence rates are stored in the database but are not used in WALIS to correct the paleo relative sea-level elevations. Future versions of WALIS should improve the fields describing land motion values. This would also entail revising the data already inserted in the database. One future addition to the database structure could be the insertion of fields reporting GIA predictions and associated metadata (e.g., information on ice and earth models) published alongside the geological records. Another possible addition is the possibility to query (and save in the database) geodatabases of GIA predictions for each site.

3) To calculate paleo RSL from the elevation of stratigraphic sea-level index points, the data compiler must insert the upper and lower limits of occurrence of a given landform or stratigraphy in the modern environment, from which the indicative meaning is calculated. These values are often not reported in the literature; therefore, in the WALIS interface, we suggest using IMCalc (Lorscheid and Rovere, 2019) to infer them from global wave and tide atlases and typical landform limits. However, as noted by the original authors of IMCalc, this tool only gives a first-order quantification of the indicative meaning, which should be replaced wherever possible by local information. We surmise that this is a challenge for the subsequent versions of WALIS and the scientific community working on



Last Interglacial sea-level at large to start reporting, alongside fossil coastal sequences, and quantitative information on their modern analogues (for an example, see Vyverberg et al., 2018).

4)  As reported in the introduction, WALIS was preceded by efforts to create a global sea-level database of Holocene data (Khan et al., 2019), called HOLSEA. This database is in spreadsheet format, but there are already visualization tools built around it, stemming from a MySQL structure (Drechsel et al., 2021). Following the interest of the sea-level community in the WALIS interface, and thanks to a "Data Stewardship Scholarship" awarded by PAGES (the Past Global Changes project) to A. Rovere and N. Khan, we started implementing the HOLSEA structure into WALIS. This work is still in "beta" version but, once completed, will allow data compilers to insert Holocene data via the WALIS interface. At the time of writing, it is possible to insert Holocene data, but the data is not yet fully implemented in the workflow shown in Figure 1. The final goal is to make a unique database including all sea-level proxies, regardless of their age.

## 6 Using WALIS

WALIS 1.0 is released under a Creative Commons Attribution license 4.0 (CC BY 4.0, https://creativecommons.org/licenses/by/4.0/). This license allows anyone to share and adapt WALIS for any use, also commercially, provided that the proper attribution is given. We encourage anyone to cite the review papers in the ESSD Special Issue alongside with the original works in any new manuscript arising from further research that builds upon the data compiled in WALIS 1.0. Credit to the database administrators, key contributors, data compilers, and funding agencies that made WALIS possible should also be given, including a suggested acknowledgment line embedded in the interface and in the WALIS 1.0 distribution.

Data in WALIS 1.0 are correct to the best of our knowledge, but we cannot exclude the presence of errors or defects. We encourage anyone finding issues or errata in WALIS 1.0 to suggest corrections using the online tools described in this manuscript or writing directly to the database administrators. Another important disclaimer is that WALIS 1.0 indicates sites (retrieved from literature) where samples were collected. There is no implied guarantee that the coordinates are accurate (some may be gathered from old maps or outdated site descriptions), that these sites are accessible, or that new samples can be lawfully collected at the same location. For the sake of the preservation of geological heritage, we encourage anyone who is planning the collection of new samples to cross-check whether the original samples reported in WALIS are still available from the original data creators to avoid collecting new material.



## 7 Conclusions

WALIS 1.0 collects data and metadata for thousands of sites globally carrying information on Last Interglacial sea-level proxies. Each data point in WALIS 1.0 is related to the complete set of information needed to describe a sea-level index point, its age constraints, and all relevant metadata associated with these properties. We built the structure of WALIS following the guidelines of Düsterhus et al. (2016), to ensure database Accessibility, Transparency, Trust, Availability, Continuity, Completeness, and Communication of content (the so-called ATTAC[3] properties). WALIS 1.0 is **accessible** in several formats, some of which are non-proprietary: SQL, CSV, XLS and GeoJSON. Data is also accessible (upon free registration) via the interface. **Transparency** is guaranteed via the interface and the documentation, which give ample details on the database structure and its fields. Also, the quality of data is addressed, as described in the sections above. WALIS includes tools to increase **trust** between data compilers and data creators, namely the requirement, for each record, to insert literature references and the reminder, set on every instance of WALIS, to credit original authors. The data and code used to export data from the database, as well as the database documentation, are **available** on open-access repositories and can be forked on GitHub for subsequent modification. We embedded within the WALIS structure tools to allow the **continuity** of updating. The open-access repository we selected for the code and data includes versioning, and modifications to the existing records can be proposed and maintained in the track record. The WALIS structure is, to the best of our knowledge, **complete** as it includes all the relevant uncertainties and metadata useful to contextualize the database information. The content of WALIS is also **communicated** via an easy-to-use and intuitive visualization interface, that can be freely accessed and for which the code is also available open-access. We foresee that WALIS will be a valuable resource to the wider paleoclimate community to facilitate data-model integration and intercomparisons, assessments of sea level reconstructions between different studies and different regions, as well as comparisons between past sea level history and other paleoclimate proxy data.

### Acknowledgments

The data used in this study were extracted from WALIS, a sea-level database interface developed by the ERC Starting Grant WARMCOASTS (ERC-StG-802414), in collaboration with PALSEA (PAGES / INQUA) working group. The database structure was designed by A. Rovere, D. Ryan, T. Lorscheid, A. Dutton, P. Chutcharavan, D. Brill, N. Jankowski, D. Mueller, M. Bartz, E. Gowan and K. Cohen. The data points used in this study were contributed to WALIS by: Roland Freisleben, Deirdre Ryan, Peter Chutcharavan, Evan Gowan, Ann-Kathrin Petersen, Ciro Cerrone, Evan Tam, Jessica Creveling and Schmitty Thompson, Daniel Muhs, Oana Alexandra Dumitru, Alessio Rovere, WALIS Admin, Alexander Simms, Kim Cohen, Patrick Boyden, Kathrine Maxwell, Nadine Hallmann, Víctor Cartelle, Karla Zurisadai Rubio Sandoval, Gilbert Camoin, Andrew Cooper, Matteo Vacchi, Alexandra Villa, Simon Steidle, Alessandro Fontana, Dorit Sivan and





Ehud Galili, Zomenia Zomeni, Rob Barnett, Muhammad Abdullah Saeed, Khan, Natasha Barlow, Barbara Mauz, Abdullah Saeed, Kai, John Doherty, Dominik Brill, Sebastian Garzon, Melanie Bartz (in order of numbers of records inserted).

**Data Availability**

The data presented in this paper is available through Zenodo: https://doi.org/10.5281/zenodo.6623428 (Rovere et al., 2022).

**Funding**

This work was funded by the European Research Council (ERC) under the European Union's Horizon 2020 research and innovation programme (grant agreement n. 802414 to AR). AD acknowledges support from the National Science Foundation grant #702740 and #2041325. The development of the WALIS visualization interface was sponsored by the PAGES "Data Stewardship Scholarship" given to Sebastian Garzòn. PAGES is supported by the Swiss Academy of Sciences and the Chinese Academy of Sciences.

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
