# Peer review of "The World Atlas of Last Interglacial Shorelines (Version 1.0)"

_Earth System Science Data, 2022_

## Author Response (AR1)

Dear Editor,

We thank you and the reviewers for providing constructive feedback on our work. Overall, the reviewers are requesting minor revisions and some more explanations to improve the clarity of the MS. Please find below a rebuttal to each comment, alongside the modifications to the original MS. We also submit a track-changes version of the original MS.

In the following, the reviewers' comments are inked in black, while our replies are highlighted in gray. If the text is highlighted in yellow, editorial input is requested.

**Rev 1 – Nicole Khan**

I enjoyed reading "The World Atlas of Last Interglacial Shorelines (Version 1.0)" by Rovere et al. The paper provides overview of preceding database efforts and the WALIS database presented in the special issue. Coming from the perspective of someone who works on Holocene databases, most of my comments and suggestions have to do with clear and consistent usage of terminology, especially for terms that were primarily developed with Holocene data in mind. I've made more specific comments on this point (and other additional minor issues) by line or figure number below:

We would like to thank Nicole Khan for assessing our MS and providing constructive feedback. We incorporated most her comments in the main text, as outlined below.

Line 18-19: Not entirely clear what is the difference between 'sea-level proxies', 'dated samples', and 'other time constraints', but perhaps it will be explained later on in the paper.

We believe this becomes clear as the reader goes on. However, we added some examples in parentheses to clarify what each of these items is. The sentence now reads:
*The database contains 4545 sea-level proxies (e.g., marine terraces or fossil beach deposits), 4110 dated samples (e.g., corals dated with U-series) and 280 other time constraints (e.g., biostratigraphic constraints or tephra layers), interconnected with several tables containing accessory data and metadata.*

Line 74: "work" not "works"
Line 75: "work" not "works"
Line 79: replace "to" with "from"
Line 83: Correct referencing format of Pedoja et al., 2014

All the above has been corrected

Line 129: "… are not duplicated in the database" Does this mean that multiple interpretations of the same sample is not allowed?

Not really. What we mean here is that we discourage duplicating very common records, such as "differential GNSS" as survey system, or "Ordnance datum" as type of sea-level datum. These are broadly used, so we think it makes sense that everyone inserting a new survey method, or datum, should be aware of what has been inserted before. However, anyone can insert a "new" datapoint for a datapoint that is already in the database. To better explain how different interpretations of the same samples are handled, we added an item in the "future developments" (section 5.3)

Line 148: "a" not "an"

Done

Line 161: "reporting on samples and their dating" - what aspect of samples?

We slightly modified the sentence, which now reads:
*The central concept is that the description of a single sea-level index point must include data and metadata describing its stratigraphy and relationship with former sea level and must be tied to a series of tables reporting on dated samples, and details on how both samples and index point have been measured.*

Line 163 (Fig 2): Unclear what is meant by "sea-level datum"

We mean "vertical datum". This has been made homogenous within the text, as we found out we used this wording in other parts of the text and figures.

Line 163: all samples need information about their geographic position (and elevation), not just index points, right? This is what the wording implies.

Yes, geographic position and elevation must be given for both samples and index points. We added a "both" in the sentence to make this explicit.

Line 164: what is specifically meant by "included" in this context?

We meant that the studies must be "referenced" rather than "included". Fixed.

Line 185: "This can happen when different sites containing sea-level index points correlate to the same sample dated, for example, with U-series." Sorry, does this sentence suggest that index points from multiple sites in the database can be generated from a single date present only at one site? To me, this would violate the traditional definition of an index point described in Shennan et al., 2015, Handbook for Sea-level Research, where the location, age, elevation and indicative meaning of a dated sample, not a dated stratigraphic unit is defined. I presume the assumption is that the age of that stratigraphic unit is known within some uncertainty from one location and then can be assumed elsewhere. What about instances where there is only one date from a stratigraphic unit, but a sample from another location/different sites in that same unit? That unit could have formed over hundreds of years to millennia, so is it likely that a single date is representative of the entire unit? Some further explanation, especially of caveats or assumptions, is needed here or elsewhere.

Thanks for this comment, we rephrased to make the concept behind the many-to-many more clear. Indeed, this is probably a relaxation of the concept of "index point" by Shennan, but it does not violate it, in our opinion. When a sample is not available in direct connection with the sea-level proxy, often the associated stratigraphy is assigned a general "MIS 5e" age. By retaining in the database not only the name/properties of the stratigraphic unit but

also the dated samples associated to that unit, we think we are giving more background information to any user on the validity of the stratigraphic correlation. We rewrote the sentence as follows, hoping it is more clear.

*It is also possible, under certain conditions, that a single dated sample can be used within various "RSL stratigraphy" records. This can happen when different sites containing sea-level index points correlate to a single stratigraphic unit dated elsewhere with, for example, U-series. In such case, the WALIS structure allows connecting one or more sea-level index points with one stratigraphic unit, but also with one or more radiometric ages that have been used to assign an age to that unit. The distance between the sea-level proxy and the closest dated sample is calculated and stored in a column of the exported spreadsheet. This kind of many-to-many relationship between sea-level proxies and dated samples allows retaining not only the age information for a given sea-level proxy, but also retaining information about how that age has been established. This is a crucial difference with both previous Last Interglacial sea-level databases (Hibbert et al., 2016; Pedoja et al., 2014; Ferranti et al., 2006). In these databases, the relationship between samples and sea-level proxies is always one-to-one.*

Line 188-189: "In these databases, the relationship between these two entities is always one-to-one (e.g., one radiocarbon sample corresponding to one peat layer indicating a former RSL)." This is not strictly true for the Holocene sea-level databases - there can be multiple radiocarbon samples from the same peat layer that indicate one or multiple positions of RSL depending on the depths of the radiocarbon samples and the data creator's interpretation. I think the examples of "many-to-many" and "one-to-one" relationships could be described more clearly in this paragraph. Furthermore, what's the benefit of allowing for one-to-many and many-to-many relationships from an administrator, data compiler or end user's perspective?

Thanks for this, we were not aware of this possibility in the Holocene database, therefore we deleted the reference to it. See last part oft he modified sentence above.

Line 192: "PHP" - should this abbreviation be defined?

It was defined (Hyptertext Preprocessor) at the very first mention.

Line 195: Define "varchar"

Done, it is an indeterminate string length data type

Line 240: "Two manuscripts for which data is available in WALIS 1.0 are only available as preprints." Which manuscripts and will they eventually be published in peer reviewed journals?

We are not sure what the plan is for those manuscripts. The authors chose not to follow through post-revision, but the data included in the databases were cross-checked by the editorial team and they are scientifically sound. We ask the editor what to do in this case, however we would propose to keep these data in. We identified the preprints in the table.

Line 275: add "chronological" before "constraints" for clarity.
Done

Line 275: "There is no upper limit on the number of constraints associated with a single sea-level proxy." This makes sense, but some description of how age uncertainties are/should be considered should be provided.

Excellent point. The new version of the visualization interface allows to do exactly this. We updated the description of the visualization interface and added this sentence at the end of this paragraph.
*The choice on how to calculate final age uncertainties for a sea-level proxy including several chronological constraints is left to the end user. However, in the data visualization interface, we propose an approach where each age within the same sea-level proxy is sampled using a Monte-Carlo approach within a random or uniform distribution (depending on the type of constraint), and a synthetic age is calculated by combining the samples available for the proxy.*

Line 276: Can an example of a maximum or minimum limiting age be given? I assume it's from stratigraphic units above (younger) or below (older) the unit from which the sea-level proxy was derived?

We provided an example within this sentence:
*This option may be chosen when, for example, only infinite radiocarbon ages are available as an age constraint for a beach deposit, indicating thus that the deposit is older than 50-60 ka, but it is not possible to assign a radiometric age.*

Line 277: Wording of sentence starting with "Together…" is a bit unclear; can you please rephrase?

We rephrased the sentence as follows, hoping it is clearer now:
*In summary, in WALIS 1.0 there are nine possible combinations (**Error! Reference source not found.**) to define a sea-level proxy (sea-level index point, marine or terrestrial limiting) and its associated age (older/younger than, or with defined age).*

Line 284: Add "types of" before "sea-level indicators"

Done, thanks for the correction

Line 311 (and Fig 9): Do you mean orthometric (instead of 'ordnance') datum? My understanding is that "Ordnance Datum" is a specific vertical datum used in the UK that measures heights relative to historic MSL recorded at specific tide gauges (Newlyn, Belfast, and Malin). Some clarity on use of terminology (i.e., providing a definition of each term in the context of the database) is important here when presenting fields of metadata in the database.

Yes, we meant Orthometric. We corrected the text and figure. Thanks for noticing our mistake. We also corrected some discrepancies between the term "sea level datum" and

"vertical datum" in the database and in the interface, adopting the latter as more correct in terms of geodetic measurements (see modifications to the database).

Line 319: "the WALIS interface requests as mandatory information the elevation of the proxy and the associated 2σ error" conflicts with what is written in Figure 10, which says "Elevation errors in WALIS are insterted but the compiler as +/- 1σ"

Thanks for noticing this. The correct notation is +/- 1 sigma. We corrected the text.

Line 332: What's a "modal limit"? Odd phrasing to me.

We substituted „limit" to „depth range". This should clarify.

Fig 1: A few questions/suggestions to improve clarity:
1. From this flow diagram, it seems at present, an end user can only access data through zenodo as .csv, .xlsx or .geoJson files if they do not have SQL knowledge. There is a 'Visualisation via Shiny App' option on the diagram, but it's unclear how an end user accesses this app from this diagram.

We modified the diagram to show how an end user can access the visualization, which is directly via a webpage.

2. There seem to be some details lacking between the data compiler and PHP interface - can there be bulk uploads via csv or excel files or can data only be added one point at a time.

At present, it is not possible to make bulk uploads directly, but only via direct contact with the database administrators. However, we put this as a possible future feature in the "future directions"

3. There are some actions (e.g., "Mod/Delete request") and objects ("Jupyter notebooks for database queries") in the diagram, but it's somewhat unclear who can perform these actions, or what the objects relate to. It might be useful to colour-code the diagram to indicate which actions are possible for the different parties involved (admin, data compiler and creator, end user) and have different symbols for actions and different states of data.

We added an arrow to clarify that mod/delete requests can be done by the data compiler or by the end user, as written in the text. We would prefer not to color code the diagram, as we feel it would add a layer of complication to the description of the structure.

Fig 2: Why do the number of records on the right hand side of the figure differ from the number on the left? They are also referred to as "records" in the text in the top right of the figure and "amount of data" in the figure caption. Some explanation and consistency in terminology would avoid confusion.

Thanks for noticing this glitch. In the right hand side, we report the number of columns per table. In the left hand side the number of records (=rows) in each table. We edited the caption to make it clearer.

Fig 3: The names of tables shown here (and described in the text) do not correspond to their names in the WALIS documentation (found here: https://walis-help.readthedocs.io/en/latest/database.html#). As a a result, it was difficult to relate the contents of the paper to the support documentation. Perhaps the tables in the support documentation can be shown here in parentheses to help.

Thanks for noticing, there was a mismatch in the labels of each table, which were updated here but not in the readthedocs. We now fixed the docs. In case the editor agrees, we can export a pdf of the readthedocs and add it here.

Fig 4: Is this the PHP interface described in Fig 1? Or is this the mySQL database? Or does this not appear on Fig 1? Should be referred to consistently (either PHP interface, MySQL database, or WALIS database interface) for clarity here and in the text.

This is the PHP interface described in Figure 1. We edited the caption to make it more clear.

Fig 7: Assuming these descriptions apply to the data visualisation tool, can the RSL and age error bars and boxes be defined in terms of the uncertainties they represent? For example, a "SLI" here shows 2 sigma RSL and age uncertainties? For the limiting type points, are the error bars plotted at the midpoint of the elevation or age uncertainty distribution of the data point or at the upper/lower end of the distribution? Also, should the y-axis be labelled as "Relative sea level" rather than "Paleo sea level"?

Done, we modified figure 7 to clarify the questions above. We think now it is clearer for the reader.

**Rev.2 - Amila Sandaruwan Ratnayake**

The World Atlas of Last Interglacial Shorelines (Version 1.0). This is an interesting topic and important for the literature. In addition, this is a well-organized study. This makes the data presented here worthy of publication. After addressing below minor typos, this study could be a nice contribution ESSD research community. The data of this study are generally interesting and the study is suitable for the readership of the wider paleoclimate community, especially in PALSEA.

We would like to thank Amila Sandaruwan Ratnayake for assessing our MS and providing constructive feedback. We incorporated most comments in the main text, as outlined below.

The scientific rationale of this work is elaborated nicely highlighting (i) what is the scientific question you are addressing? (ii) and how you will do it? (iii) why is it important for international readers to care about your manuscript? (iv) what is the novelty of your manuscript? In addition, the authors can add an additional sentence to specifically mention the objective of the study.

We think that this is already implicit in the last two sentences of the introduction. Keeping in mind that this is an editorial, collating several previously published manuscripts, the aim is really to describe the database, as we wrote there.

"Discussion" is critically discussed. In the Discussion, the author interpreted the results with proper explanation. Results are critically analyzed, and discussion and synthesis parts are also well organized. Argumentation and interpretation of your data are consistent. As a reader, I loved reading your interesting paper.

Thank you very much for this. We are happy you liked our work.

Specific comments
Line 83: Pedoja et al., 2014. Use a consist format like Pedoja et al. (2014). Consider this comment hereafter such as in line 384, 433, etc.

Thank you. We went through the MS and fixed these instances.

Lines 113-117: Add an additional sentence to highlight the difference between "Data compilers" and "Database administrators".

We think that this is pretty clear from the description given in the text.

Line 168: It is better to add a coastal geomorphologists as well. I feel that they are a powerful subgroup for shoreline changes.

Thanks. Added

Lines: 188-189 Use chronological or alphabetical order for the references. Consider this comment hereafter such as lines 425, 438, 442, etc.

We are not sure if these should go in alphabetical vs chronological order. The citation style zenodo suggests for ESSD seems to make them alphabetical. Asking the editor if it is OK to check this in the copyediting phase.

Figure 4: Is there any possibility to increase the quality of screenshot in Figure 4?

Unfortunately, no. We tried, but this is what we can get. Sorry about this. However, these tools are accessible so it should not be a problem to check them out in their webpage.

**Rev.3 - Georgia Grant**

This database represents significant effort by the authors to provide a user-friendly interface that considers multiple access methods for those who wish to add data or export data. Particularly, I'm impressed by the number of fields included which allows comprehensive data analysis and review. This database is of significant value to the community and importantly, allows for future development. I congratulate the authors on

including an evaluation metric, this is of considerable value. The database structure and content is very well explained.

We would like to thank Georgia Grant for assessing our MS and providing constructive feedback. We incorporated most comments in the main text, as outlined below.

The authors give appropriate acknowledgment to previous databases from which this has added to and developed from, and have clearly made significant effort to work with the community to include all available data.

We believe this is crucial to ensure that credit is given where credit is due. Thanks for highlighting this.

The statement starting line 97 which outlines the differences in previous compilations to WALIS would be useful earlier in Section 2, and could be added in Setcion 1, to highlight the contribution of WALIS upfront.

While we agree with this suggestion, we also think that a section on its own (as the one we wrote) gives more visibility to previous compilations. We are not sure how to proceed here. So, the question goes to the editor: do we get rid of Section 2 and include it within section 1, or should we keep it as it is?

Further to this point, while the manuscripts referenced in Table 2 outline the compiled data in WALIS 1.0, the following line 252 references data in WALIS 'spanning more than a century'. Can you clarify that these datasets have been included in the submissions of the Special Issue referenced in the table (if that is correct).

Yes, thanks for noticing this. Indeed, the 2130 references are those from which data and metadata were extracted by the database compilers. We changed the sentence as follows: *The papers listed in **Error! Reference source not found.** contain data and metadata standardized from 2130 references, spanning more than one century of published scientific literature.*

Phrases referencing standardization are inconsistent and therefore unclear as to whether the database applies standardization to the data itself or standardizes the approach to reporting. I infer from the manuscript, it is taken to mean standardization approach and think this should be clarified when used. Further to this point, on line 38, it would be useful when mentioning 'approaches to standardize', more explaination on what these were and how WALIS addresses these would be insightful.

We went through the MS keeping in mind this comment. As a result, we made some small changes, including some to Paragraph 2 and 3 in the introduction, trying to clarify what we mean by standardization. We hope this addresses the reviewer's comment.

It would be useful to have a definition of terms as a table or list to clarify the use of e.g. sea-level index points, sea-level reference points, sea-level indicators and sea-level proxies.

Figure 2 is of significant use for this purpose, and could be combined with Fig 3 as a second panel to show the relationships. This is well-discussed until line 267.

We would prefer to keep these two figures separated. As per the definition of terms or table list, we are not sure if this paper is the right place for this kind of table. There is already a recent version of the handbook of sea level research (cited in the MS) that clarifies all these terms, so we would refrain from duplicating efforts. We ask the editor whether we should do this, or if reference to the literature is OK in this case.

In introducing the LIG relevance (paragraph line 56), more basic information on the importance of this period would be welcome. In particular, it would be useful for the reader to have clear examples of how sea-level change/ reconstructing sea-level has contributed to broader understanding of the climate system/ sensitivity/ processes. This speaks directly to the usefulness of this database.

In the second-to-last paragraph of the review we added the following sentence, which we believe gives enough context to the reader to understand the usefulness of such a database. *"In MIS 5e, climate was slightly warmer than pre-industrial (with global mean atmospheric temperatures 1-2°C above pre-industrial, McKay et al., 2011), probably as a consequence of greater insolation at high latitudes. In such warmer climate, global mean sea level was up to 8-10 meters higher than today (Dutton and Lambeck, 2012; Dutton et al., 2015; Dyer et al., 2021). Last Interglacial global mean sea-level estimates have been used as benchmarks for ice models projecting future sea-level changes in face of global warming (DeConto et al., 2021). Gilford et al., 2020 highlighted that improving field measurements of MIS 5e sea levels might help improving the accuracy of future sea-level projections."*

Move Fig. 1 further up the manuscript - a very uesful figure.

Thank you. We put Figure 1 here as it is where the different parts of the platform are described. However, we leave to the copyeditor the option to bring it up in the MS in the final version if this is OK with journal guidelines (e.g. appearing slightly before being mentioned in the text). Fine by us.

Fig. 7 is a fantastic visualization of the limits of a sea-level index point. Does the shading represent the theoretical uncertainty/confidence? If so, please state.

We slightly modified Figure 7 according to Rev. 1 comments, and decided to get rid of the shading. It was basically a graphical display to give an idea where the probability for the RSL at a certain point in time would be higher, but it was not quantitative, so probably poorly informative. We hope the reviewer agrees with our decision.

Line 52, starting 'These efforts ...' could use sentence restructure.

Done

Line 195 term 'varchar'  needs definition.

Done, it is an indeterminate string length data type

I particularly enjoyed the discussion on geographic gaps, which shows the breadth of research and understanding of the authors on this topic.

Thank you! Lots of work to do, but exciting way forward!

**Corrections to the database (from 1.0-review to 1.0)**

- Updated description of the sldatum table from "sea level datum" to "vertical datum"
- In the "General_Type" column, the value "ordnance or geodetic datum" was updated to "orthometric or geodetic datum
- Corrections of minor typos in the "vertical datum" table